# Postoperative Circulating Tumor DNA Can Predict High Risk Patients with Colorectal Cancer Based on Next-Generation Sequencing

**DOI:** 10.3390/cancers13164190

**Published:** 2021-08-20

**Authors:** Chul Seung Lee, Hoon Seok Kim, Jeoffrey Schageman, In Kyu Lee, Myungshin Kim, Yonggoo Kim

**Affiliations:** 1Department of Surgery, Seoul St. Mary’s Hospital, College of Medicine, The Catholic University of Korea, Seoul 06591, Korea; lcs0610@hanmail.net; 2Department of Laboratory Medicine, Seoul St. Mary’s Hospital, College of Medicine, The Catholic University of Korea, Seoul 06591, Korea; hskim11@catholic.ac.kr (H.S.K.); yonggoo@catholic.ac.kr (Y.K.); 3Catholic Genetic Laboratory Center, Seoul St. Mary’s Hospital, College of Medicine, The Catholic University of Korea, Seoul 06591, Korea; 4Thermo Fisher Scientific, Austin, TX 78701, USA; Jeoffrey.Schageman@thermofisher.com

**Keywords:** circulating tumor DNA, colorectal cancer, next-generation sequencing

## Abstract

**Simple Summary:**

Circulating tumor DNA (ctDNA) is a minimally invasive biomarker useful for monitoring minimum residual disease, recurrence, and treatment response in colorectal cancer (CRC). We analyzed circulating tumor DNA from patients with CRC to evaluate analytical and clinical performances using next-generation sequencing (NGS). It is clear that postoperative circulating tumor DNA detection provides valuable information to determine whether a patient might at high risk of disease recurrence or have a persistent tumor lesion. The NGS assay not only showed excellent analytical performance, but also shows a state-of-art diagnostic option in patient-oriented precision medicine.

**Abstract:**

The objective of this study was to characterize circulating tumor DNA (ctDNA) mutations in colorectal cancer (CRC) patients and evaluate their prognostic values during treatment. Forty-nine patients with CRC planned for operation were enrolled. A total of 115 plasma samples were collected pre-operation, post-operation, and post-chemotherapy. ctDNA analysis was performed using next-generation sequencing (NGS) including 14 genes. In 22 (44.9%) out of 49 patients, at least one mutation (40 total mutations) was detected in the initial plasma sample. The median sum of variant allele frequency was 0.74% (range: 0.10–29.57%). *TP53* mutations were the most frequent (17 of 49 patients, 34.7%), followed by *APC* (18.4%), *KRAS* (12.2%), *FBXW7* (8.2%), *NRAS* (2.0%), *PIK3CA* (2.0%), and *SMAD4* (2.0%). After surgery, five (14.3%) out of 35 patients harbored ctDNA mutation. All five patients experienced relapse or metastasis during follow-up. It was noteworthy that all three patients with persistent ctDNA relapsed after R0 resection. After chemotherapy, ctDNA analysis was performed for 31 patients, all of which were ctDNA-negative. Analytical and clinical performances of NGS to utilize ctDNA in CRC were determined. Results revealed that postoperative ctDNA might serve as a marker for identifying risk of recurrence, thus contributing to patient-oriented treatment strategies.

## 1. Introduction

Circulating tumor DNA (ctDNA), known as tumor-derived cell-free DNA (cfDNA), can be detected in the acellular part of peripheral blood from cancer patients [1]. Since ctDNA contains tumor-specific mutations that represent tumor nature and status, studies on ctDNA have been conducted for various cancer types [2,3,4,5,6,7,8]. Likewise, the potential of ctDNA in colorectal cancer (CRC) as a minimally invasive biomarker has been highlighted in many recent studies for treatment response, prognosis prediction, minimum residual disease (MRD), and recurrence monitoring [9,10,11,12,13,14,15,16,17,18]. ctDNA profiling can effectively reveal the genomic landscape of cancer compared to tissue profiling by reflecting heterogeneity and providing biologically essential findings including therapeutic resistance [9]. This is useful for rapidly identifying resistance-causing mutations and for selecting therapeutic agents [10,11,12,13]. In addition, mutations observed at diagnosis can be used as MRD markers to evaluate therapeutic response and predict patient prognosis related to recurrence and survival, thus allowing appropriate timing for ctDNA testing before and/or after treatment [14,15,16,17,18]. Serial ctDNA levels in CRC-resected patients can be used to detect disease recurrence earlier than conventional postoperative surveillance [19]. There are several ongoing randomized trials worldwide to determine the clinical utility of ctDNA in colorectal cancer [20].

High analytical sensitivity is the most essential requirement for MRD evaluation. Traditional DNA analysis methods such as Sanger sequencing are not sensitive enough to detect somatic mutations of ctDNA in plasma [21]. Capture-based next generation sequencing (NGS) enables the enrichment of genomic regions of interest by hybridizing target genes to antisense oligonucleotides prior to sequencing [22]. In addition, further advances in error-correcting technique allow for the detection of mutations with a very high sensitivity by distinguishing the background artifacts [23].

The aim of this study was to determine the usefulness of the NGS assay for analyzing ctDNA in CRC, focusing on the presence of ctDNA before surgery, after surgery, and after chemotherapy. ctDNA associated with clinicopathologic findings was characterized. Analytical and clinical performances of the NGS assay were evaluated to define the optimal method and time point for predicting high risk patients.

## 2. Materials and Methods

### 2.1. Evaluating Analytical Performance

AcroMetrix™ Oncology Hotspot Control (Thermo Fisher Scientific, Waltham, MA, USA) was used to evaluate the analytical performance including sensitivity and specificity. Six thousand copies of 0.1% and 0.5% fragmented control mixtures were prepared where controls were diluted into the background of fragmented genomic DNA. Eight and sixteen replicates were carried out to confirm the sensitivity and specificity at 0.5% and 0.1% limit of detection (LOD), respectively.

In addition, LODs were validated using the Multiplex I Cell Free DNA Reference Standard Set (Horizon Discovery, Cambridge, UK). The reference material covered multiple engineered single nucleotide variants (SNVs) with four mutations at 5%, 1%, and 0.1% allelic frequencies in genes including *KRAS*, *NRAS*, and *PIK3CA*. The reference materials were analyzed as described in the section ‘Amplicon library preparation, sequencing, and data analysis’ below.

### 2.2. Patients

Between January 2018 and December 2019, a total of 49 patients with CRC were enrolled. We enrolled patients over time during that period in a real clinical setting without any specific exclusion criteria. The median age of the patient cohort was 64 years (range: 43–87 years). There were 23 men and 26 women. A surgeon determined the operative method for each case considering the patient’s medical status and tumor location. All patients followed our institution’s enhanced postoperative recovery (ERAS) protocol, which included colorectal surgery [24]. Conventional multiport laparoscopic or robotic technique was used for minimal invasive surgery. Of the 49 patients, 44 received complete (R0) oncological resections and adequate nodal harvest followed by adjuvant chemotherapy including 5-fluorouracil, leucovorin, and oxaliplatin (FOLFOX) regimen. The other five patients received R1 or R2 resection by distant metastasis or resection margin involvement. They received palliative therapy including bevacizumab or cetuximab plus standard chemotherapy (FOLFOX or FOLFIRI; folinic acid, 5-fluorouracil, and irinotecan) [25]. 

### 2.3. Sample Collection and cfDNA Extraction

Peripheral venous blood samples (*n* = 115) were collected for patients at three time points: two days before surgery (pre-op), 10 days after surgery (post-op), and at the end day of the last chemotherapy (post-chemo). At least 20 mL of blood was taken into EDTA-containing tubes. Plasma was separated within 4 h after sample collection. Obtained plasma was centrifuged at 2000 *g* for 5 min and at 16,000 *g* for 10 min, immediately aliquoted, and stored at −80 °C. cfDNA was isolated from 4 mL of plasma using a MagMAX Cell-Free DNA Isolation Kit (Applied Biosystems, Foster City, CA, USA) with a KingFisher Duo Prime Magnetic Particle Processor (Thermo Fisher Scientific, Waltham, MA, USA), according to each manufacturer’s instructions. The concentration of the purified plasma cfDNA was measured using a Qubit 2.0 fluorometer (Thermo Fisher Scientific, Waltham, MA, USA) in combination with a Qubit dsDNA HS Assay Kit (Thermo Fisher Scientific, Waltham, MA, USA) according to the manufacturer’s instructions.

### 2.4. Amplicon Library Preparation, Sequencing, and Data Analysis

Oncomine™ Colon cfDNA Assay (Thermo Fisher Scientific, Waltham, MA, USA) was used to generate libraries from cfDNA following the manufacturer’s instructions. The NGS panel covering 14 genes (*AKT1*, *BRAF*, *CTNNB1*, *EGFR*, *ERBB2*, *FBXW7*, *GNAS*, *KRAS*, *MAP2K1*, *NRAS*, *PIK3CA*, *SMAD4*, *TP53*, and *APC*) with >240 hot spots (SNVs and short Indels) was used in this study. Quality control of libraries was performed using an Ion Library TaqMan^®^ Quantitation Kit (Applied Biosystems, Foster City, CA, USA) on an ABI 7500 Real-Time PCR system (Applied Biosystems) and a TapeStation D1000 Kit (Agilent Technologies). Six-plex library pool was applied on an Ion 530 chip. Ion Chef™ System and Ion 530™ Kit-Chef were used for template preparation, followed by sequencing on an Ion S5 XL Sequencer according to the manufacturer’s instructions (Thermo Fisher Scientific, Waltham, MA, USA).

Sequence data were processed for primary and secondary analyses using the standard Ion Torrent Suite Software running on a Torrent Server. Raw signal data were analyzed using Torrent Suite v 5.10.1 and Ion Reporter (Thermo Fisher Scientific, Waltham, MA, USA). The pipeline included signal processing, base calling, quality score assignment, adapter trimming, PCR duplicate removal, read alignment, quality control of mapping quality, coverage analysis, and variant calling. Sequenced reads were aligned against the UCSC hg19 reference genome (Genome Reference Consortium GRCh37). Sequence variants were identified using the Ion Reporter software v5.10 and Ion AmpliSeq HD Workflow template for Liquid Biopsy–w1.4–DNA–Single Sample (Thermo Fisher Scientific, Waltham, MA, USA). The coverage of each amplicon was determined using the Coverage Analysis Plugin software v.5.10.0 (Thermo Fisher Scientific, Waltham, MA, USA). The application of UMIs enabled the grouping of reads into molecular families. Random errors generated during the library construction and sequencing process were removed automatically.

### 2.5. Statistical Analysis

Differences between groups were evaluated using the Chi-squared or Fisher’s exact test for categorical variables and Mann–Whitney (rank sum) test or Student’s *t*-test for continuous variables where appropriate. The prognostic significance of covariates affecting disease recurrence was determined by Cox proportional hazards regression model. Variables assigned as significant prognostic factors in univariate analysis were included in multivariate analysis. Disease free survival (DFS) curves were analyzed using the Kaplan–Meier method and compared by the log-rank test for univariate analysis. All statistical analyses were performed using SPSS software version 24.0 (IBM SPSS Statistics^®^, Armonk, NY, USA).

## 3. Results

### 3.1. Analytical Performance of Oncomine™ Colon cfDNA Assay and Quality Control Matrices

Analytical performance was evaluated using 0.1% and 0.5% fragmented control mixtures. Sensitivities of 0.1% and 0.5% LOD were 85.9% and 100%, respectively. Specificities of 0.1% and 0.5% LOD were both 100%.

The median concentration of plasma cfDNA from 115 clinical samples was 11.3 ng/mL (range: 2.9–53.7 ng/mL). The median sequencing read coverage was 51,584 (range: 6754–133,847). All samples achieved a median read coverage greater than 25,000. The median molecular coverage was 3066 (range: 258–8679). GC contents did not influence the mean amplicon read depth (R^2^ = 0.33, Appendix A).

### 3.2. Patient Characteristics and Pretreatment ctDNA Detection

Patients’ baseline clinicopathologic characteristics and their associations with ctDNA status are shown in Table 1. Twenty-two (44.9%) out of 49 patients harbored pre-op ctDNA mutations. Among them, 20 (90.9%) patients were classified to higher T stage (T3 and T4, *p* = 0.006). Interestingly, vascular and perineural invasion were more frequently observed in patients with pre-op ctDNA (40.9% vs. 3.7%, *p* = 0.003; 22.7% vs. 3.7%, *p* = 0.043) while lymphatic invasions were not associated with pre-op ctDNA. In addition, higher levels of initial CEA, post-op CEA, and ascites CEA were more frequently observed in patients with pre-op ctDNA (*p* = 0.019, *p* = 0.031, and *p* = 0.031, respectively).

A total of 40 mutations were observed in 22 patients. The median number of detected ctDNA mutations was 1.5 (range: 1–5). *TP53* mutations (total 18 mutations in 17 patients, 17/49 = 34.7%) were the most commonly observed, followed by *APC* (*n* = 9, 18.4%), *KRAS* (*n* = 6, 12.2%), *FBXW7* (*n* = 4, 8.2%), *NRAS* (*n* = 1, 2.0%), *PIK3CA* (*n* = 1, 2.0%), and *SMAD4* (*n* = 1, 2.0%) (Figure 1 and Appendix A). Among the 18 *TP53* mutations, mutations in codon 248 were predominant (*n* = 7), followed by those in codon 175 (*n* = 3), 282 (*n* = 3), and 273 (*n* = 2). Patients with *TP53* mutation were associated with higher T stage, higher levels of pre-op CEA, and higher post-op CEA (*p* = 0.018, *p* = 0.002, and *p* = 0.044, respectively). All *APC* mutations were truncation mutations. The majority (78%, 7/9) occurred in the mutational cluster region (MCR). Patients with *APC* mutation were associated with higher T and TNM stage (TNM3 and TNM4), higher levels of pre-op CA19-9 and post-op CEA, and large intestinal obstruction (*p* = 0.045, *p* = 0.045, *p* = 0.035, *p* = 0.028, and *p* = 0.016, respectively). *KRAS* mutations involved G12 and G13. The significance of mutant burden was analyzed. The variant allele frequency (VAF) of each mutation was variable (median 0.94%, range: 0.06–27.48%). The sum of VAFs (VAFsum) in each sample was then calculated and the value was defined as the VAFsum in the sample. The median VAFsum was 0.74% (range: 0.10–29.57%). Patients with higher T stage, higher TNM stage, vascular invasion, and perineural invasion showed higher VAFsum (*p* = 0.015, *p* = 0.012, *p* = 0.001, and *p* = 0.019, respectively). Pre-op cfDNA concentration was found to be significantly higher in patients with higher T stage and *TP53* mutation (*p* = 0.045 and *p* = 0.033, respectively).

### 3.3. Monitoring Postoperative Recurrence

After surgery, 5 (14.3%) of 35 patients revealed ctDNA mutations. A total of 12 mutations were observed. Frequencies were as follows: *TP53* (*n* = 5, 14.3%), *APC* (*n* = 3, 8.6%), *KRAS* (*n* = 2, 5.7%), *FBXW7* (*n* = 1, 2.9%), and *SMAD4* (*n* = 1, 2.9%) (Figure 1). Vascular invasion and perineural invasion were observed more frequently in patients with persistent post-op ctDNA (80.0% vs. 13.3%, *p* = 0.001; 80.0% vs. 6.7%, *p* < 0.001). Medians of VAF and VAFsum were 2.83% (range: 0.11–9.32%) and 6.11% (range: 0.54–27.57%), respectively. Post-op VAFsum was significantly higher in patients with higher M stage (median: 0.88% vs. 0%, *p* < 0.001), vascular invasion (median: 0.27% vs. 0%, *p* = 0.002), and perineural invasion (median: 0.71% vs. 0%, *p* = 0.001).

Among the 44 patients who received R0 resections, six (8.8%) patients developed disease recurrence during follow-up. The median (range) time to recurrence was 9.9 (2.1–20.6) months and the median follow-up time for all patients was 31.4 (1.0–36.1) months. Higher M stage, perineural invasions, and post-op ctDNA status were significantly associated with disease recurrence (*p* = 0.030, *p* = 0.009, and *p* = 0.005, respectively) (Table 2). After multivariable adjustment, post-op ctDNA status remained an independent predictor of disease recurrence (*p* = 0.004) (Table 2). In 32 patients with available postoperative samples, patients with post-op ctDNA positive showed a significantly shorter DFS than patients with post-op ctDNA negative (Hazard ratio = 2.80, 95% CI: 1.68–3.92, *p* < 0.001) (Appendix A). Among the five patients who could not receive R0 resection, two with available post-op ctDNA analysis showed persistent mutations. Altogether, we found that persistent post-op ctDNA reflected the persistence of residual disease or the risk of disease recurrence.

ctDNA analyses were performed for 31 patients after chemotherapy. All available samples including two from patients with persistent post-op ctDNA showed negative results in post-chemo ctDNA analysis. Post-chemo cfDNA concentration was higher in patients with higher TNM stage (mean: 17.1 ng/mL vs. 9.3 ng/mL, *p* = 0.049) and lymphatic invasion (median 17.0 ng/mL vs. 8.7 ng/mL, *p* = 0.011).

## 4. Discussion

In this study, we evaluated pre-op, post-op, and post-chemo ctDNA mutations in CRC patients using the NGS assay. This assay revealed excellent analytical performance of sensitivity and specificity for ctDNA analysis. About 45% of CRC patients showed pre-op ctDNA mutations. In addition, patients with detectable ctDNA mutation at the initial showed higher T stage, more vascular invasion, and perineural invasion. They also showed higher CEA levels of blood and peritoneal fluid than patients without initial ctDNA mutation. These results were in line with previous studies showing that the tumor stage appeared to have a major impact on the detection rate of ctDNA mutation because tumor necrosis and apoptosis occurred more frequently in larger tumor masses [26]. Perivascular and perineural invasions were due to expanding tumor mass, which could be associated with initial ctDNA mutation in our cohort [17,27].

The distribution of ctDNA mutation was similar to previous studies [28,29]. *TP53* was the most commonly mutated gene, followed by *APC*, *KRAS*, and *FBXW7*. The *TP53* mutation frequency was 43.28% in the IARC *TP53* Database (http://www-p53.iarc.fr/, accessed on 26 April 2021) [28]. *TP53* codons 248, 273, 175, 245, 282, and 249 were the top six mutated codons, with mutational rates of 6.79%, 6.55%, 4.8%, 3.12%, 2.59% and 2.59%, respectively [29]. We also found that codons 248, 282, 175, and 273 were recurrently mutated in our cohort. They were significantly correlated with vascular and perineural invasions. These results were in line with previous studies demonstrating that *TP53* mutation was associated with a higher aggressiveness nature such as vascular and perineural invasions [30,31,32]. Inactivating mutations in the *APC* gene have been reported in 34–70% of sporadic colorectal cancer patients [33,34,35]. Studies in Korean CRC patients showed similar prevalence of *APC* mutations [36,37,38]. We also detected nine *APC* mutations that generated truncated gene products. APC inactivation is thought to be an early event in the development of CRC. It may play a pivotal role in the initiation of the adenoma–carcinoma pathway. Although *APC* is the most frequently mutated, known driver gene in CRC, its prognostic impact has not been well clarified. In recent years, studies have demonstrated the predictive importance of *APC* gene mutations for molecular targeted therapeutics [39,40]. In addition, determination of the *KRAS* mutation status is mandatory for treatment with anti-EGFR monoclonal antibodies in patients with metastatic CRC [41,42]. The frequency of *FBXW7* mutation in the present study was 8.2%. This value is consistent with that in previous studies, which reported that 10% of patients with CRC have *FBXW7* mutations [43]. *FBXW7* is a potential tumor suppressor, and mutations in the gene are thought to impair cyclin E degradation resulting in uncontrolled cell division and growth, thus resulting in cancer progression [44]. Previous studies suggested that the *FBXW7* mutation was significantly associated with shorter overall survival in CRC patients [45,46]. Based on these results, we could consider that NGS is the most efficient method to analyze ctDNA in CRC due to its simultaneous detection of various mutations in multiple genes.

Because 30−50% of CRC patients face recurrence after R0 surgery [47,48], it is essential to monitor residual disease using the optimal method at the optimal time. An early study has demonstrated that ctDNA could identify a recurrence 2–15 months (average: 10 months) earlier [14]. Another study has introduced quantitative criteria that suggest a VAF cut-off of 0.046% to predict 3-year relapse free survival for patients treated and not treated with adjuvant chemotherapy [49]. In this study, we found that the prevalence of ctDNA mutation was markedly decreased in 14.3% after surgery. Patients who showed clearance of pre-op ctDNA and maintained the status after chemotherapy did not experience disease recurrence. It was notable that we could select three patients with persistent ctDNA at 10 days after R0 resection who were facing disease recurrence. We also observed persistent post-op ctDNA who did not receive R0 resection and presumed to have remaining tumor. Taken together, these data provide evidence for the validity of post-op ctDNA for stratifying the disease status as well as the risk of recurrence. These data also suggest a need of additional therapeutic approach in those patients on the base of disease monitoring by ctDNA analyses [16,50].

Although this is the first prospective study to evaluate the clinical significance of ctDNA in CRC of Korean patients, it has several limitations. First, we did not access several post-chemo ctDNA because this study was performed in a real clinical setting. Second, the number of each clinicopathologic status was small. Therefore, we did not fully analyze the correlations between ctDNA mutation and the molecular effect that impacted the tissue nor study the response to treatment for each biomarker. Other studies have provided new ideas for treatment with respect to expression differences and response-related mechanisms that we could not perform [51]. Informative biomarkers are useful in that treatment strategies can be improved by modifying the sequence, dose, and combination of radiation therapy, chemotherapy, and surgical resection [52]. Third, the follow-up period was relatively short. Thus, we could not evaluate the association of ctDNA with long-term outcome after additional target therapy. Further studies are needed to clarify the incidence and meaning of post-chemo ctDNA as well as post-op ctDNA in a larger cohort at multiple time points.

## 5. Conclusions

The NGS assay is an adequate method to analyze ctDNA in CRC patients because it not only shows excellent analytical performance, but also technical improvement for the standardization of the assay. Higher prevalence of *TP53* and *APC* gene mutations also potentiated the necessity of the NGS assay to cover various mutations in multiple genes. It was clear that post-op ctDNA detection provides valuable information to determine whether a patient might at a high risk of disease recurrence or have a persistent tumor lesion. In addition, patients in a disease-free state maintained negative conversion status during the follow-up. Serial monitoring is needed to minimize the waiting time to treatment failure and the opportunity for intervention.

## Figures and Tables

**Figure 1 cancers-13-04190-f001:**
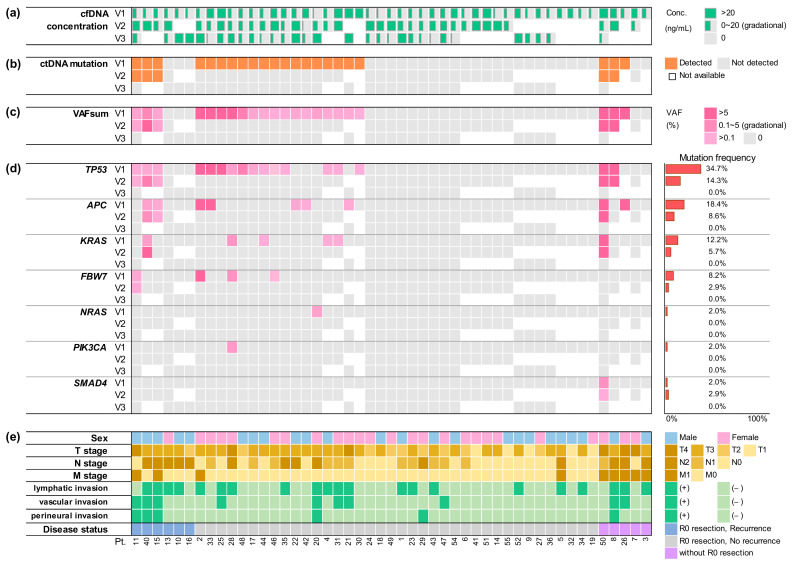
Summary of clinical characteristics and serial cfDNA analysis for 49 colorectal cancer patients. (**a**) Concentrations of cfDNA is shown gradationally by the width of green bars. (**b**) Detection of ctDNA mutation at three time points is indicated in orange. (**c**) VAFsum is shown gradationally in darker shades of pink. (**d**) VAF of mutations is displayed individually for each gene. Mutation frequencies are also provided on the right side. (**e**) Clinical and pathological characteristics is summarized by color and shade as indicated on the right; cfDNA, cell-free DNA; ctDNA, circulating tumor DNA; VAF, variant allele frequency; VAFsum, sum of VAFs; V1, pre-operation; V2, post-operation; V3, post-chemotherapy; Conc, concentration; Pt, patient number.

**Table 1 cancers-13-04190-t001:** Clinicopathologic characteristics of 49 patients with colorectal cancer.

Variable	Pre-op ctDNA Status		
	Positive	Negative	*p*
Patient No.(%)	22(44.9)	27(55.1)	
Age, years	64.9 ± 13.2	64 ± 9.7	0.774
Gender, No. of male(%)	10(45.5)	14(51.9)	0.656
T stage			0.006
pT1–2	2(9.1)	12(44.4)	
pT3–4	20(90.9)	15(55.6)	
N stage			0.056
pN0	7(31.8)	16(59.3)	
pN1–2	15(68.2)	11(40.7)	
M stage			0.44
No	17(77.3)	24(88.9)	
Yes	5(22.7)	3(11.1)	
TNM stage			0.02
pTNM1–2	5(22.7)	15(55.6)	
pTNM3–4	17(77.3)	12(44.4)	
Vascular invasion			0.003
No	13(59.1)	26(96.3)	
Yes	9(40.9)	1(3.7)	
Neural invasion			0.043
No	17(77.3)	26(96.3)	
Yes	5(22.7)	1(3.7)	
Lymphatic invasion			0.136
No	10(45.5)	18(66.7)	
Yes	12(54.5)	9(33.3)	
Microsatellite instability (+)			0.713
No	17(77.3)	23(85.2)	
Yes	5(22.7)	4(14.8)	
Peritoneal fluid cytology (+)			0.181
No	5(22.7)	11(40.7)	
Yes	17(77.3)	16(59.3)	
Initial CEA level			0.019
≤3 ng/dl	9(40.9)	20(74.1)	
>3 ng/dl	13(59.1)	7(25.9)	
Post-op CEA level			0.031
≤3 ng/dl	14(66.7)	25(92.6)	
>3 ng/dl	7(33.3)	2(7.4)	
Peritoneal fluid CEA			0.031
≤3 ng/dl	1(6.2)	6(42.9)	
>3 ng/dl	15(93.7)	8(57.1)	

ctDNA, circulating tumor DNA; CEA, carcinoembryonic antigen; Pre-op, two days before surgery; Post-op, 10 days after surgery.

**Table 2 cancers-13-04190-t002:** Univariate and multivariate regression analyses for disease recurrence in patients with colorectal cancer.

Variable	Univariate Analysis			Multivariate Analysis		
	HR	(95% CI)	*p*	HR	(95% CI)	*p*
Sex, female	0.2	(0–1.6)	0.121			
T3–4 stage	2.7	(0.3–25.7)	0.386			
N1–2 stage	5.9	(0.6–55.4)	0.121			
M1 stage	18.0	(1.3–245.6)	0.030	3.2	(0.1–152.3)	0.551
TNM3–4 stage	13.7	(0.7–260.8)	0.082			
Vascular invasion	6.4	(1–41)	0.050			
Neural invasion	17.5	(2.1–149.2)	0.009	3.9	(0.2–77)	0.368
Lymphatic invasion	3.3	(0.5–20.3)	0.201			
Microsatellite instability (+)	0.9	(0.1–8.5)	0.895			
Peritoneal fluid cytology (+)	2.7	(0.3–25.7)	0.386			
Initial CEA level >3 ng/dL	2.4	(0.4–13.6)	0.335			
Post-op CEA level >3 ng/dL	0.5	(0–9.3)	0.608			
Peritoneal fluid CEA >3 ng/dL	3.8	(0.2–80.7)	0.395			
Pre-op ctDNA (+)	1.5	(0.3–8.3)	0.664			
Post-op ctDNA (+)	133.0	(4.5–3936.6)	0.005	81.0	(4.0–1655.8)	0.004

ctDNA, circulating tumor DNA; CEA, carcinoembryonic antigen; Pre-op, two days before surgery; Post-op, 10 days after surgery.

## Data Availability

The data that support the findings of this study are available from the corresponding author upon reasonable request.

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
