# Peer review of "Postoperative Circulating Tumor DNA Can Predict High Risk Patients with Colorectal Cancer Based on Next-Generation Sequencing"

_cancers, 2021, doi:10.3390/cancers13164190_

Round 1
Reviewer 1 Report
The objective of the authors was to characterize circulating tumor DNA (ctDNA) mutations in colorectal cancer (CRC) patients and evaluate their prognostic values during treatment
the authors use a standardized and adequate method of analysis on a patient cohort and corresponding plasma samples. although the correlations with histomorphological parameters, DFS and DS are adequate, the discussion disappoints in content. There is no critical correlation on the value of the mutations found with other published evidence on the subject: - eg. what impact do ctDNA mutations have on what happens in the tissue? - no relationship has been established between the mutations found and what molecular effects they can induce in the pathways involved; - do the mutations have effects on some checkpoints highlighted in other cases on the subject? (e.g. doi: 10.2174 / 0929867326666190507084839. doi: 10.18632 / oncotarget.493Author Response
Please see the attachment.

Reviewer 2 Report
This manuscript describes the application of the Oncomine colon cfDNA assay in a cohort of 49 patients with colon cancer. The authors tested the blood of these patients pre-op, post-op and post-chemotherapy. The main finding was that the detection of cfDNA in patients post-op was associated with recurrence post Ro resection. This is a well written paper with possible implications for the use of cfDNA detection in the management of colon cancer. The main limitation is the small number of patients with only 3 patients who had cfDNA detected post-op and had recurrence. However, the authors have discussed this limitation.
I have a few comments and questions for the authors:
Line 87. Was there any inclusion or exclusion criteria in patient selection?
Lines 75-79/148-149. It is unclear how the analytical performance of the NGS assay was analyzed. Limit of detection is the lowest concentration of analyte that can be distinguished from the limit of blank. It is not clear how the sensitivity and specificity of the LOD can be derived. From the description in the methods it seems the authors diluted the control DNA in normal DNA at 0.5% and 0.1% and performed replicates of this diluted control at 8X and 16X. Usually for analytical sensitivity, serial dilution would be performed and the hit rate of the replicates would be given in a chart or table. Did the authors perform replicates of normal genomic DNA to determine the analytical specificity or lack of false positive cfDNA detection?
Table 2. What variables were adjusted in the multivariate analysis? Could this be described in the methods? For post-op cfDNA detection, was adjusting for neural invasion in multivariate analysis performed?
Line 238. There is no discussion on FBXW7 in colon cancer.
Round 2
Reviewer 2 Report
The authors have adequately addressed all the comments.